# Deep Learning–Based Brain Computed Tomography Image Classification with Hyperparameter Optimization through Transfer Learning for Stroke

**DOI:** 10.3390/diagnostics12040807

**Published:** 2022-03-25

**Authors:** Yung-Ting Chen, Yao-Liang Chen, Yi-Yun Chen, Yu-Ting Huang, Ho-Fai Wong, Jiun-Lin Yan, Jiun-Jie Wang

**Affiliations:** 1Department of Diagnostic Radiology, Chang Gung Memorial Hospital, Keelung 204201, Taiwan; yungting12@cgmh.org.tw (Y.-T.C.); rsc8418@cgmh.org.tw (Y.-Y.C.); m7131@cgmh.org.tw (Y.-T.H.); jwang@mail.cgu.edu.tw (J.-J.W.); 2Department of Medical Imaging and Intervention, Chang Gung Memorial Hospital, Chang Gung University, Linkou 333423, Taiwan; hfwong@cgmh.org.tw; 3Department of Neurosurgery, Chang Gung Memorial Hospital, Keelung 204201, Taiwan; colorgenie@cgmh.org.tw

**Keywords:** machine learning, neuroradiology, computed tomography, stroke, classification

## Abstract

Brain computed tomography (CT) is commonly used for evaluating the cerebral condition, but immediately and accurately interpreting emergent brain CT images is tedious, even for skilled neuroradiologists. Deep learning networks are commonly employed for medical image analysis because they enable efficient computer-aided diagnosis. This study proposed the use of convolutional neural network (CNN)-based deep learning models for efficient classification of strokes based on unenhanced brain CT image findings into normal, hemorrhage, infarction, and other categories. The included CNN models were CNN-2, VGG-16, and ResNet-50, all of which were pretrained through transfer learning with various data sizes, mini-batch sizes, and optimizers. Their performance in classifying unenhanced brain CT images was tested thereafter. This performance was then compared with the outcomes in other studies on deep learning–based hemorrhagic or ischemic stroke diagnoses. The results revealed that among our CNN-2, VGG-16, and ResNet-50 analyzed by considering several hyperparameters and environments, the CNN-2 and ResNet-50 outperformed the VGG-16, with an accuracy of 0.9872; however, ResNet-50 required a longer time to present the outcome than did the other networks. Moreover, our models performed much better than those reported previously. In conclusion, after appropriate hyperparameter optimization, our deep learning–based models can be applied to clinical scenarios where neurologist or radiologist may need to verify whether their patients have a hemorrhage stroke, an infarction, and any other symptom.

## 1. Introduction

Brain computed tomography (CT) is a modality most commonly used for evaluating the cerebral condition [1]. It is more widely available, fast, and cost-effective than is brain magnetic resonance imaging. Although brain CT was developed in the 1970s, its widespread clinical use became achievable only recently, after the introduction of rapid, large-coverage multidetector-row CT scanners. Key clinical applications for brain CT include the diagnoses of cerebral hemorrhage and ischemia neoplasm and evaluation of the mass effect after hemorrhage, neoplasm, and cerebral edema secondary to ischemia. However, immediate and highly accurate interpretation of emergent CT images remains time-consuming and laborious, even for skilled neuroradiologists [2]. Lodwick described computer-aided diagnosis (CAD) for the first time. Since then, a wide variety of lesion detection systems have been reported [3,4]. The usefulness of CAD depends on the number of true- and false-positive markers. High-performance CAD systems are appreciated by radiologists in the screening practices [5]. For nodule detection in chest X-ray, CAD can even outperform the diagnosis efficiency of the unexperienced radiologists [6]. At present, some CAD systems have received approval from the U.S. Food and Drug Administration [7,8]. Compared with traditional CAD (which may be limited by detecting specific disease and requiring a work-alone station), deep learning can handle more complicated condition. With a relatively wide scope, deep learning delivers multiple answers. The software improvements over the last few decades have enabled not only a considerable amount of research on image processing algorithms and methodologies but also rapid, faultless identification and quantification of abnormalities in scanned regions [9].

Deep learning, a well-employed network for medicine [10,11,12], can outperform humans in diagnosis. Li et al. [13] proposed a U-net based model to identify cerebral hemorrhage, which has many advantages over human expertise, but it demands much manpower and time for segmentation. We aimed at developing a simple model, like a CNN-based system, to classify the results of brain CT as red-dot systems. Training a conventional convolutional neural network (CNN) from scratch typically requires a considerable amount of data. Nevertheless, through transfer learning, a small amount of data can become sufficient for finetuning a pretrained model [14]. For patients with preexisting cerebral changes, a final human check of the deep learning–based diagnosis is required to ensure credibility; nevertheless, it can improve the clinical decision-making of neuroradiologists [15]. Owing to the wide variety commercial models available, understanding the mechanism underlying the “black box” in computer operation is difficult. Moreover, studies that defined a reference to hyperparameter adjustment in the various models in certain conditions have been limited.

In this study, pretrained models including CNN-2, VGG-16, and ResNet-50—with varied data sizes, mini-batch sizes, and optimizers—were compared in terms of their performance in classifying unenhanced brain CT image findings into normal, acute or subacute hemorrhage, acute infarction, and other categories. We also reviewed other studies on deep learning–based diagnoses of hemorrhagic and ischemic stroke and compared their outcome with ours.

## 2. Materials and Methods

### 2.1. Data Preparation

#### 2.1.1. Data Collection

This retrospective study was approved by our institutional review board, which also waived the requirement for obtaining patient informed consent and using anonymized patient imaging data. Our dataset included 24,769 unenhanced brain CT images from 1715 patients collected over 1 July–1 October 2019. Of these images, 4382, 6102, 3860, and 2995 were from healthy patients, patients with hemorrhage, patients with infarction, and patients with other findings, respectively. The uneven ratio between each group also reflects the patient structure in our district. Moreover, only 10% of these images (*n* = 2476) constituted the testing dataset (Table 1), which was used to evaluate the objective performance to ensure that the accuracy of the data for a specific period was not represented. We labelled these images based on clinical results after at least a half-year follow up and split them randomly into training dataset, validation dataset, and testing dataset, based on the case-based level. Once the abnormal findings were included in case-based images, the results were defined as abnormal and further classified into hemorrhage, infarct, or others.

All images were confirmed by two radiologists, who had 1 year and 20 years of experience, to be ground truth for deep learning, with the other findings including craniotomy, deep brain implantation, severe motion, or artifacts.

#### 2.1.2. Preprocessing Steps

The original images from the dataset were denominated and cropped to a standard size of 224 × 224 × 3 using MicroDicom, a multiphase algorithm used for image processing.

#### 2.1.3. Data Normalization

Data normalization is an important step for numerical stabilization of a CNN; it changes the range of the pixel intensity values and converts an input image into a range of pixel values that is more familiar or normal to the senses. It is faster and more stable for gradient descent [16]. In the current study, the pixel value of input images was downsampled to 224 × 224 × 3 through simply scaling with centering in the range of 0–1 for each channel.

#### 2.1.4. Data Augmentation

CNNs are data-hungry and can be more powerful and perform better with larger datasets [17]. However, collecting medical data can be difficult. Data augmentation, in which a series of transformation procedures are used while preserving the ordinary labels, has widely been applied to multiply the numbers of the images. This adjustment can also aid in preventing overfitting as a regularizer. We randomly applied each horizontal flipping, rotation, shift, zoom, and shear for each image. So, the range of our data size is about 1- to 2-fold of the original (24,767 × 1~24,767 × 2). Figure 1, Figure 2 and Figure 3 display the layout of our proposed neural network architecture.

#### 2.1.5. Transfer Learning

Deep learning is data-dependent, which is the most challenging problem of this modality. Here, sufficient training with a large amount of information is required for the network to recognize data patterns. Ideally, both training and testing data are assumed to have same distribution and features. However, constructing sufficiently large datasets can be laborious, time-consuming, and even impossible in some cases. In this condition, transfer learning from one pretrained domain to another is beneficial and efficient [14,18].

### 2.2. Hyperparameters for CNN-Based Models

In this study, the included CNN models were the pretrained CNN-2, VGG-16, and ResNet-50 from ImageNet [19,20] (Table 2). They were tuned with the following hyperparameters: learning rate = 0.0001, max epochs = 50, and mini-batch sizes = 8, 16, 32, 64, and 128. Early stopping was applied if gradient exploding occurred. The dropout rates were 0.3 and 0.5 in all included models. ResNet-50 was applied without dropout for further comparison. The optimizers were respectively added with Adam [21], SGD [22], and RMSProp [23]. We applied a rectified linear unit (ReLU) activation function [24], converting the input weighted sum into the node’s output, in each convolutional layer [25]. ReLU was implanted in the hidden layers of the CNN. We also divided our dataset sizes into four categories: <1000, 1000–5000, 5000–9000, and >10,000; these were compared in the best environment in each model. We want to compare the different amounts of dataset sizes in different models with the best environment.

### 2.3. Hardware

All processing was performed on Intel Xeon Gold 6126 3.7GHz CPU, DDR4-2666 256GB memory, and NVIDIA Tesla V100 PCIe 32GB GPU using Python (Tensorflow, Keras, matplotlib).

### 2.4. Performance Evaluation

We evaluated our models in terms of the accuracy, precision, recall, and F1-score of the proposed CNN and other pretrained models and plotted curves for calculating the areas under the curves (AUCs), with epochs on X axis and accuracy on Y axis.

## 3. Results

### 3.1. Comparison of the Validation Accuracy among the CNN Models Trained with Different Mini-Batch Sizes

With mini-batch sizes of 8, 16, 32, 64, and 128, the validation accuracy of CNN-2 was 0.9787, 0.9829, 0.9840, 0.9808, and 0.9872, respectively; that of VGG-16 was 00.8480, 0.8948, 0.9543, 0.9575, and 0.9479, respectively; that of ResNet-50 with dropout was 0.9606, 0.9659, 0.9734, 0.9681, and 0.9638, respectively; and that of ResNet-50 without dropout was 0.9770, 0.9851, 0.9776, 0.9808, and 0.9872, respectively. Thus, the highest accuracy of CNN-2, VGG-16, ResNet-50 with dropout, and ResNet-50 without dropout was noted at mini-batch sizes of 128, 64, 32, and 128, respectively (Table 3). Hence, these models were selected for comparing the performance with different optimizers.

### 3.2. Training Performance

The training performance data, including training loss, validation loss, and validation accuracy, obtained by the selected models at different epochs are listed in Table 4 and illustrated in Figure 4. Moreover, Figure 5 presents the confusion matrices for all the models with testing data.

For infarction, hemorrhage, normal, and other findings, the CNN-2 demonstrated a precision of 0.99, 1.0, 0.98, and 0.98, respectively; a recall rate of 0.99, 0.98, 1, and 0.98, respectively; and an F1 score of 0.99, 0.99, 0.99, and 0.98, respectively; moreover, the overall accuracy and AUC of the CNN-2 were 0.9872 and 0.98, respectively. For infarction, hemorrhage, normal, and other findings, the VGG-16 demonstrated a precision of 0.95, 0.98, 0.97, and 0.97, respectively; a recall rate of 0.97, 0.97, 0.95, and 0.92, respectively; and an F1 score of 0.96, 0.98, 0.96, and 0.94, respectively; moreover, the overall accuracy and AUC of the VGG-16 were 0.9575 and 0.97, respectively. For infarction, hemorrhage, normal, and other findings, the ResNet-50 demonstrated a precision of 0.99, 1.0, 0.98, and 0.98, respectively; a recall rate of 0.98, 0.98, 1.0, and 0.99, respectively; and an F1 score of 0.99, 0.99, 0.99, and 0.99, respectively; moreover, the overall accuracy and AUC of the ResNet-50 were 0.9872 and 0.99, respectively. Finally, the total training time for CNN-2, VGG-16, and ResNet-50 was 8 min 19 s, 1 min 58 s, and 27 min 53 s, respectively (Table 5).

### 3.3. Comparison of Different Optimization Methods

The Adam optimizer was used for training all the models. To evaluate the classification effectiveness of this optimization method, the results were compared with those of other efficient optimization methods, such as SGD and RMSProp, as shown in Table 6. The Adam optimizer was found to provide the most efficient results.

### 3.4. Comparison of Different Data Sizes for Training

Training data sizes of <1000, 1000–5000, 5000–9000, and >10,000, respectively, provided a validation accuracy of 0.3312, 0.6111, 0.8655, and 0.9872 in the CNN-2; 0.5250, 0.7075, 0.8884, and 0.9575 in the VGG-16; and 0.5750, 0.7266, 0.9567, and 0.9872 in the ResNet-50 (Table 7). In general, the accuracy of all the included models increased in proportion to the data sizes, with the highest accuracy with data sizes of >10,000. Nevertheless, the accuracy values of the ResNet-50 for data sizes of 5000–9000 and >10,000 were very close, indicating that high accuracy (≥0.9567) can be achieved by using ResNet-50 even for a data size >5000. This is in contrast to the results for the CNN-2 and VGG-16, which could not achieve high accuracy for a data size <10,000.

## 4. Discussion

On the basis of the current results, CNN-based deep learning models can be used to detect strokes automatically and with high accuracy after hyperparameter optimization. We also compared the performance with different hyperparameters, regularizers, and data sizes.

Hemorrhagic stroke, a common and fatal disease, often presents symptoms similar to the more commonly diagnosed ischemic stroke. However, the treatment for hemorrhagic stroke is focused on controlling the bleeding from ruptured cerebral vasculatures or aneurysms, whereas that for ischemic stroke is focused on recannulating clot blockages in cerebral arteries. Misdiagnosis leading to the erroneous use of anticoagulant agents for treating a hemorrhagic stroke can cause death. Unenhanced brain CT is the most common and recommended test of choice to identify the two stroke types. Nevertheless, it is more difficult to make a diagnosis of subtle infarct only based on unenhanced CT. Schriger et al. [26] claimed that in the absence of support from a radiologist, the accuracy of this interpretation is only 0.67 among emergency physicians treating patients with stroke.

Since the advent of deep learning, the use of brain CT images for accurate prediction of critical anomalies has received considerable attention. Several attempts have thus been made to develop a reliable diagnosis model using deep learning methods. Transfer learning has also been extensively used with the recent CNN-based networks [27]. However, most of these methods have employed an imbalanced and limited amount of data, which has led to unsatisfactory results. In the current study, we developed a system that classifies hemorrhagic and ischemic strokes by using numerous brain CT images sampled uniformly from a patient population.

Here, we comprehensively evaluated the effectiveness of the three most efficient CNN models, namely CNN-2, VGG-16, and ResNet-50, in the classification of hemorrhagic and ischemic strokes from brain CT images after hyperparameter optimization. One of the most crucial hyperparameters considered in this study was the mini-batch size. We accordingly identified the mini-batch size that provided the highest validation accuracy for the CNN-2, VGG-16, and ResNet-50 with regard to classifying the brain CT images. Moreover, the best performing models in this study were found to be CNN-2 and ResNet-50 (highest accuracy = 0.9872). Grewal et al. [28] developed an automatic intracranial hemorrhage detection model based on deep learning, with a sensitivity of 0.8864 and a precision of 0.8124 in a dataset of 77 brain CT images interpreted by three radiologists. However, the authors included a small dataset and detected only hemorrhagic stroke in their analysis. Moreover, Prevedello et al. [29] assessed the performance of a deep learning algorithm to detect hemorrhage, mass effect, hydrocephalus, and suspected acute infarction by using a dataset of 50 brain CT images and reported AUCs of 0.91 for hemorrhage, mass effect, and hydrocephalus and only 0.81 for suspected acute infarction. In the current study, after optimization, all three models, trained with relatively more data, demonstrated outstanding performance, with F1 scores >0.95.

In addition to accuracy, efficiency is an important factor in medical image classification. In this study, the VGG-16 and CNN-2 required only about 2 and 8 min on average to provide the outcome, respectively, which is nearly 14 and 4 times faster than the time taken by the ResNet-50, respectively. We thus believe that this significant difference in time consumption occurs because of the relatively complicated structure of ResNet-50, with numerous hidden layers. If the data size is bigger, it costs several folds of time higher than ours, and the difference in time consumption is bigger.

The images that are false positive are illustrated in Figure 6. However, we cannot provide a clear explanation of why the classification failed, due to the mechanism underlying the “black box”. There is no relationship between size, laterality, location, or augmentation process. It is possible that an increase in data size will achieve better performance.

Tandel et al. [30] reported that the simplest CNN-1 could classify benign and malignant gliomas with high efficiency from magnetic resonance images, with further high-efficiency subclassification into low- and high-grade malignant gliomas enabled by the CNN-2. Further segmentation of low-grade and high-grade malignant gliomas can be performed using the CNN-3 and CNN-4. They utilized artificial neural networks (ANNs) as feature extraction algorithms and a CNN as classifier, with high accuracy of 0.98. Despite the simple architecture of a CNN, it is effective in classifying gliomas from magnetic resonance images; hence, we considered it efficient to classify the four categories in our study.

Ioffe and Szegedy [31] claimed that removing dropout as an optimizer from ResNet allows the network to achieve increased accuracy. We noted similar results for our brain CT images (Table 3), even in different mini-batch sizes.

Activation function is key in deep learning architectures, and many types of nonlinear activation functions exist. Pedamonti [24] reported that both ReLU and LeakyReLU are suitable activation functions for CNN-based models, particularly deeper neural networks. In the current study, we also compared the different activation functions on VGG-16 for stroke classification, and no apparent difference between ReLU and LeakyReLU was noted; nevertheless, their performance was better than that of the sigmoid activation function. However, considering the wide range of activation functions available, including ReLU, LeakyReLU, ELU, SELU, and sigmoid, the activation function most suitable for classifying stroke images warrants further investigation.

The clinical application of our result is mainly as a classifier for radiologists, who can quickly issue a warning to clinicians. Rather than a “red-dot system”, we will apply this result to further develop a system combined with the nature language process (NLP). Although understanding the mechanism underlying the “black box” in computer operation is difficult, we can supply many image inputs for NLP training through this classifier. We are committed to doing this in the future.

## 5. Limitations

Although deep learning has a considerable potential in medical applications, some related limitations include data availability and variability. At the institutional level, the contrast, noise, and resolution levels used for CT vary, and this can impede adaptation for deep learning. Moreover, data privacy is essential when considering the use of medical images for research and development, and this limits the amount of data available. Data generalization is achievable through transfer learning and data augmentation (both of which can produce additional features to learn), such that any related problem may be resolved within one day; however, this may take a long time and a considerable amount of available data to accomplish.

## 6. Conclusions

In this study, the use of CNN-based deep learning was proposed for efficient classification of hemorrhagic and ischemic stroke using unenhanced brain CT images. The CNN models CNN-2, VGG-16, and ResNet-50, pretrained through transfer learning, were analyzed by considering several hyperparameters and environments, and their results were compared. CNN-2 and ResNet-50 outperformed the VGG-16 with an accuracy of 0.9872; however, ResNet-50 required longer time than the other networks. After optimization, the tested models may be applied by radiologists to verify their screening results and thus reduce their workload. Our results also pave the way for further development of effective deep CNN models (using residual connections) for increasing the diagnosis accuracy for stroke. In the future, we will verify the effectiveness of our proposed models in terms of time required and performance and explore the use of optimization algorithms along with the models used in this study to design a more reliable model.

## Figures and Tables

**Figure 1 diagnostics-12-00807-f001:**
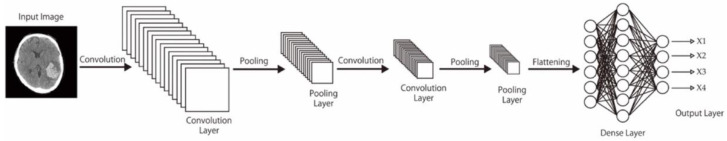
CNN-2 model. The CNN-2 model consists of 2 convolution layers, each followed by a pooling layer and a dense layer. A dense layer is also referred to as a fully connected (FC) layer (illustrated in Figure 2 and Figure 3). Our dense layers consist of 3 hidden layers, including 50,176 input units and 4 output units. Dense layers of VGG-16 and CNN-2 are equipped with rectified linear unit (ReLU) activation function, but the dense layer of Resnet-50 is not.

**Figure 2 diagnostics-12-00807-f002:**
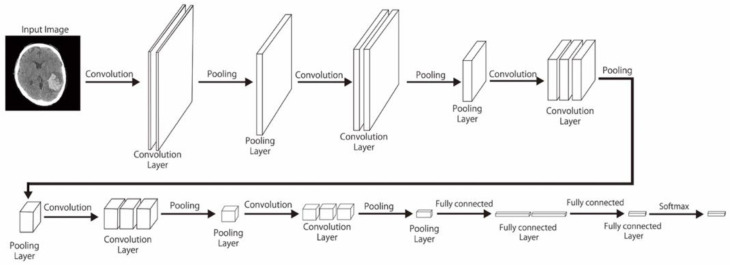
VGG-16 model. Illustration of using the VGG-16 for transfer learning. The convolution layers can be used as a feature extractor, and the fully connected layers can be trained as a classifier.

**Figure 3 diagnostics-12-00807-f003:**
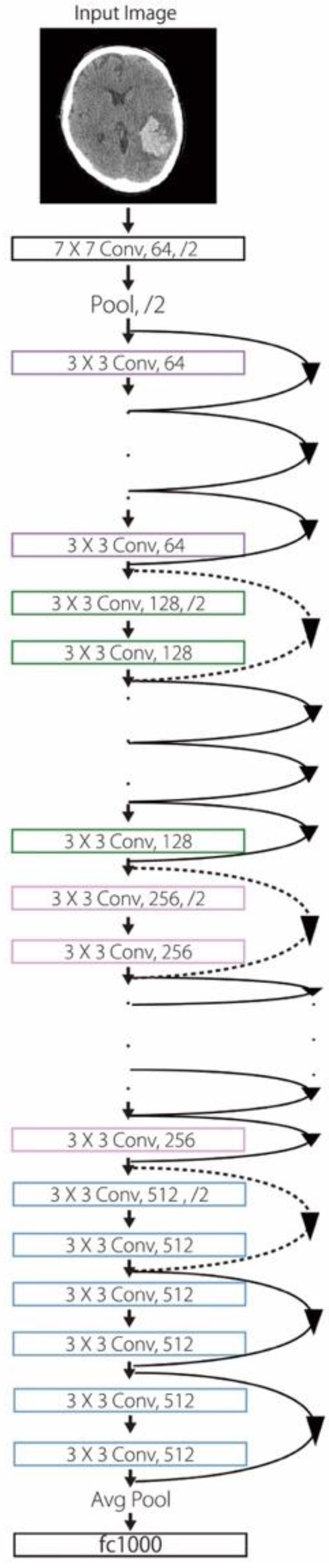
ResNet-50 model. A residual network with 50 parameter layers. The dotted shortcuts increase dimensions.

**Figure 4 diagnostics-12-00807-f004:**
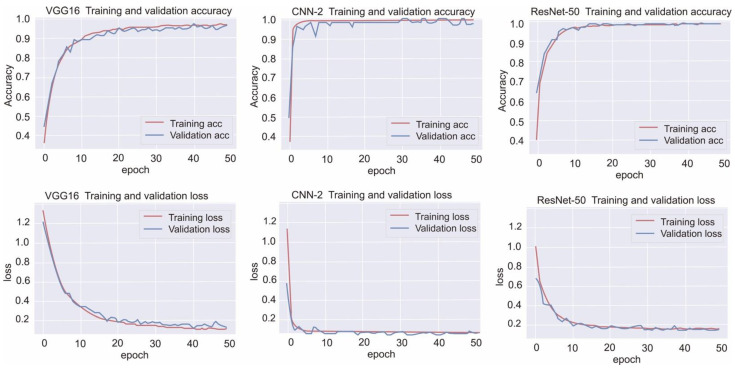
Training performance data of the selected VGG-16, CNN-2, and ResNet-50 models.

**Figure 5 diagnostics-12-00807-f005:**
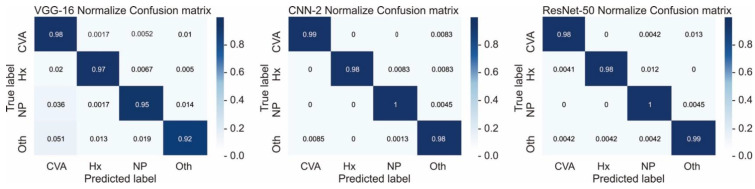
Confusion matrices of the selected VGG-16, CNN-2, and ResNet-50 models. CVA = infarction, Hx = hemorrhage, NP = normal, Oth = others.

**Figure 6 diagnostics-12-00807-f006:**
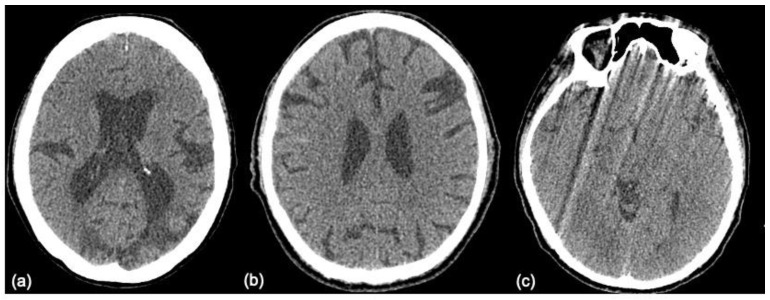
Examples of the false positive images: (**a**) CVA is misinterpreted as Hx, (**b**) Hx is misinterpreted as CVA, and (**c**) Oth is misinterpreted as CVA. CVA = infarction, Hx = hemorrhage, NP = normal, Oth = others.

**Table 1 diagnostics-12-00807-t001:** Dataset before augmentation.

Class	Dataset (Images)		
	Train	Validation	Test
Normal	4382	1252	626
Hemorrhage	6102	1743	872
Infarct	3860	1103	551
Others	2995	856	427
Total (1715 people/24,769 images)	17,339 (70%)	4954 (20%)	2476 (10%)

**Table 2 diagnostics-12-00807-t002:** Architectural descriptions of the pretrained CNN models used in this study.

Model	Layers	Parameters	Input Layer Size	Output Layer Size
**CNN-2**	2	11,954,982/11,954,982	(224,224,3)	(4,1)
**VGG-16**	16	2,514,154/2,514,154	(224,224,3)	(4,1)
**Resnet-50**	50	1,313,796/24,901,508	(224,224,3)	(4,1)

**Table 3 diagnostics-12-00807-t003:** Validation accuracy of the CNN models trained with different mini-batch sizes.

Model	Batch Size				
	8	16	32	64	128
CNN-2	0.9787	0.9829	0.9840	0.9808	0.9872
VGG-16	0.8480	0.8948	0.9543	0.9575	0.9479
Resnet-50	0.9606	0.9659	0.9734	0.9681	0.9638
Resnet-50 without dropout	0.9770	0.9851	0.9776	0.9808	0.9872

**Table 4 diagnostics-12-00807-t004:** Training performance data at different epochs.

Model	Epoch	Train Loss	Valid Loss	Valid Accuracy
CNN-2	1	1.2124	0.4175	0.8619
	…	…	…	…
	49	0.0127	0.0554	0.9858
	50	0.0125	0.0528	0.9858
VGG-16	1	1.3249	1.2072	0.4463
	…	…	…	…
	49	0.0842	0.1094	0.9596
	50	0.0886	0.1002	0.9660
Resnet-50	1	0.9963	0.6531	0.7559
	…	…	…	…
	49	0.0154	0.0067	0.9932
	50	0.0205	0.0179	0.9941

**Table 5 diagnostics-12-00807-t005:** Best performance in the testing groups with different models. The overall accuracy of CNN-2, VGG-16 and Resnet-50 are 0.9872, 0.9575 and 0.9872, respectively. The area under curve (AUC) of CNN-2, VGG-16 and Resnet-50 are 0.98, 0.97 and 0.99, respectively. The total training time of CNN-2, VGG-16 and Resnet-50 are 8 min 19.76 s, 1 min 58.91 s and 27 min 53.60 s.

**CNN-2**	**Precision**	**Recall**	**F1-Score**
Infarct	0.99	0.99	0.99
Hemorrhage	1.00	0.98	0.99
Normal	0.98	1.00	0.99
Other	0.98	0.98	0.98
**VGG16**	**Precision**	**Recall**	**F1-Score**
Infarct	0.95	0.97	0.96
Hemorrhage	0.98	0.97	0.98
Normal	0.97	0.95	0.96
Other	0.97	0.92	0.94
**Resnet-50**	**Precision**	**Recall**	**F1-Score**
Infarct	0.99	0.98	0.99
Hemorrhage	1.00	0.98	0.99
Normal	0.98	1.00	0.99
Other	0.98	0.99	0.99

**Table 6 diagnostics-12-00807-t006:** Performance of the CNN models trained with different optimization methods.

Model	Optimizer	Precision	Sensitivity	Accuracy	F1-Score
CNN-2	SGD	0.98	0.98	0.986	0.98
	Adam	0.98	0.98	0.984	0.98
	RMSProp	0.98	0.98	0.985	0.98
VGG-16	SGD	0.82	0.45	0.447	0.57
	Adam	0.96	0.95	0.945	0.95
	RMSProp	0.94	0.93	0.931	0.94
Resnet-50	SGD	0.73	0.73	0.732	0.73
	Adam	0.98	0.98	0.977	0.98
	RMSProp	0.98	0.98	0.981	0.98

**Table 7 diagnostics-12-00807-t007:** Accuracy obtained by the CNN models trained with different data sizes.

Model	Data Size			
	<1000	1000–5000	5000–9000	>10,000
CNN-2	0.3312	0.6111	0.8655	0.9872
VGG-16	0.5250	0.7075	0.8884	0.9575
Resnet-50	0.5750	0.7266	0.9567	0.9872

## Data Availability

Not applicable.

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
