# Peer review of "Deep Learning–Based Brain Computed Tomography Image Classification with Hyperparameter Optimization through Transfer Learning for Stroke"

_diagnostics, 2022, doi:10.3390/diagnostics12040807_

Round 1

Reviewer 1 Report

This paper uses CNN-based deep learning to classify stroke CT images.

I have some questions and comments:

1) Line 53 – 55: You have jumped from U-net to CNN. Please explain your relationship. ....

2) L95: "Data normalization of the .... "???????

3) L106: "We applied horizontal flipping, rotation, shift, zoom, and shear for each image." Does it mean that your image collection is 5 times larger than the original?

In Table 1 you have 24767 images, does it mean you have used 24767*5 images?

4) L134:"We also divided our dataset sizes into four categories: <1000, 1000–5000, 5000–9000 and >10000; these were compared in the best environment in each model. ".Why? What does it mean? It is possible to read an explanation in L196 but not before. You have tested different training sets....It is well known that in deep learning the more data you have the better.... Do you want to show the minimum amount of data to achieve good scores? Or what kind of CNN needs less data for good training? Why is that important if you have enough data?

5) Figure 4: Maybe, does VGG16 need some extra epochs to find the stability of the diagnostic curves?

6) After using CNN-2, VGG-16 or Restnet-50, do you use the same dense layer?, Please describe the dense layer architecture

7) Please provide an image for each category, infarct, hemorrhage and normal . After classification, could you provide an image that is a false positive and/or a false negative? In this case, could you explain why the classification of that image has failed?

8) L226: "In the current study, we developed a system that classifies hemorrhagic and ischemic strokes by using numerous brain CT images sampled uni-formly from a patient population." Why do you think your data is well balanced? In Table 1, the number of hemorrhage images is twice the number of "others."

9) L246: do you mean the time a predictive model needs to learn? Or maybe the time a predictive model needs to classify an image? If it is the time you need to learn, I think it is not important because you only need to learn once. On the other hand, if it is a time to classify an image, 2 minutes seems a long time to me.

Author Response

Response to Reviewer 1 Comments

Point 1:  Line 53 – 55: You have jumped from U-net to CNN. Please explain your relationship. ....

Response 1:  Li et al. [9] proposed a U-net based model to identify cerebral hemorrhage, which has many advantages over human expertise, but it costs lots of manpower and time for segmentation. We wonder if we can develop a simple model, like CNN, to classify the results of brain CT as red-dot systems.

Point 2:  L95: "Data normalization of the .... "???????

Response 2:  Data normalization of the is an important step for numerical stabilization of a CNN.

Point 3:  L106: "We applied horizontal flipping, rotation, shift, zoom, and shear for each image." Does it mean that your image collection is 5 times larger than the original?

In Table 1 you have 24767 images, does it mean you have used 24767*5 images?

Response 3:  We randomly applied each type of augmentation for each image. So, the range of our data size is about 1 to 2 folds of the original (24767*1~24767*2).

Point 4:  L134:"We also divided our dataset sizes into four categories: <1000, 1000–5000, 5000–9000 and >10000; these were compared in the best environment in each model. ".Why? What does it mean? It is possible to read an explanation in L196 but not before. You have tested different training sets....It is well known that in deep learning the more data you have the better.... Do you want to show the minimum amount of data to achieve good scores? Or what kind of CNN needs less data for good training? Why is that important if you have enough data?

Response 4:  Yes, we want to show the minimum amount of data to achieve good scores.

As a beginner in the areas of artificial intelligence, they will hesitate for the insufficient amounts of data. So we want to compare the different amounts of data in different model with the best en-vironment. In different to CNN-2 or VGG-16, high accuracy value of Resnet-50 can be achieved with the data size of 5,000-9,000. Afterall, the larger the volume of data, the better the performance did.

Point 5:  Figure 4: Maybe, does VGG16 need some extra epochs to find the stability of the diagnostic curves?

Response 5:  We think the validation accuracy is stable with the numerical value as the table below. Generally, although each epoch on VGG-16 takes more time to complete,it needs less epoch to reach a certain training accuracy than Resnet-50.

Epoch

42

43

44

45

46

47

48

49

50

Accuracy

0.9580

0.9576

0.9557

0.9610

0.9572

0.9437

0.9551

0.9596

0.9660

Point 6:  After using CNN-2, VGG-16 or Restnet-50, do you use the same dense layer?, Please describe the dense layer architecture

Response 6: A dense layer is also referred to as a fully-connected layer (illustrated in Fig.1 to Fig. 3). Our dense layers consisit of 3 hidden layers, including 50,176 input units and 4 output units. Dense layers of VGG-16 and CNN-2 are equipped with rectified linear unit (ReLU) activation function, but the dense layer of Resnet-50 didn’t.

Point 7:  Please provide an image for each category, infarct, hemorrhage and normal. After classification, could you provide an image that is a false positive and/or a false negative? In this case, could you explain why the classification of that image has failed?

Response 7:  Understanding the mechanism underlying the “black-box” in computer operation is difficult. Although we can provide some images that is false positive, we cannot explain why the classification has failed. There is no relationship between size, laterality, location or augmentation process. We think all we can do is to increase the data size to reach the better performance as possible.

Point 8:  L226: "In the current study, we developed a system that classifies hemorrhagic and ischemic strokes by using numerous brain CT images sampled uniformly from a patient population." Why do you think your data is well balanced? In Table 1, the number of hemorrhage images is twice the number of "others."

Response 8: Although our data seems not balanced, it is collected from 1715 patients over July 1–October 1, 2019 in our institution. The uneven ratio between each group also reflects the patient structure in our district. Like we want to find out the minimum amount of traning group, inhomogeneity between each groups can also be a important factor for machine learning. However, it doesn’t causes an impact in our study for classifying.

Point 9:  L246: do you mean the time a predictive model needs to learn? Or maybe the time a predictive model needs to classify an image? If it is the time you need to learn, I think it is not important because you only need to learn once. On the other hand, if it is a time to classify an image, 2 minutes seems a long time to me.

Response 9: Yes, it means the time a predictive model needs to learn.

We want to present every details about training a model, because not much study has done it before. And it can be a reference about time consumption of CNN-2, VGG-16 and Resnet-50 when the data size is about 25,000 images. If the data size is bigger, it costs several folds of time higher than ours.

Reviewer 2 Report

This is study using convolutional neural network (CNN)-based deep learning models for classification of strokes based on unenhanced brain CT image findings into normal, hemorrhage, infarction, and other categories. The results showed that among our CNN-2, VGG-16, and ResNet-50 analyzed by considering several hyperparameters and environments, the CNN-2 and ResNet-50 outperformed the VGG-16, with an accuracy of 0.9872. The following points need revisions.

  1. The manuscript needs more references to support the background introduction and discussion. For example, line 31-37, line 44-51, line 221-225 quote no citations. For such kind of paper, 26 references are not enough.
  2. The study was approved by the institution. The IRB number should be included.
  3. This is a study based on CT images, not MRI images. However, the authors described “…CNN-1 could classify benign and malignant gliomas with high efficiency from magnetic resonance images….we considered it efficient to classify the four brain condition categories in our study” (line 252-258). This should be explained.
  4. There is lack of discussion about the clinical application of this result. In the clinical settings, for patients with intracranial hemorrhage, the causes may be traumatic or spontaneous. Can these models apply to the differentiation of traumatic intracerebral hemorrhage or subarachnoid hemorrhage from hemorrhagic stroke? Can this study differentiate the perifocal edema of brain tumors from ischemic stroke? How can we use these models for any brain CT at emergency room? These critical points should be thoroughly discussed.
  5. Many sentences need further English editing (some short paragraphs should be merged.) to reconstruct the manuscript.

Author Response

Response to Reviewer 2 Comments

Point 1:  The manuscript needs more references to support the background introduction and discussion. For example, line 31-37, line 44-51, line 221-225 quote no citations. For such kind of paper, 26 references are not enough.

Response 1:  Additional reference are replenished in introduction and discussion.

Point 2:  The study was approved by the institution. The IRB number should be included.

Response 2:  The study was approved by the Ethics Committee of Keelung branch of Chang Gung Memorial Hospital, Taiwan (grant number 201801762B0C101, 201801762B0C102) and was conducted according to the Declaration of Helsinki.

Point 3:  This is a study based on CT images, not MRI images. However, the authors described “…CNN-1 could classify benign and malignant gliomas with high efficiency from magnetic resonance images….we considered it efficient to classify the four brain condition categories in our study” (line 252-258). This should be explained.

Response 3:  In the study of Tandel et al., they utilize artificial neural network (ANN) as feature extraction algorithms. The performance was good with the accuracy of 0.98. However, the ANN-based system costs lots of manpower and time for features extraction, wavelet features and features selection. But, as being a classifier for hemorrhage or infraction, CNN-based system is enough. For our study, it isn’t such complicated feature extraction and manifestation as classifying brain tumor.

Point 4:  There is lack of discussion about the clinical application of this result. In the clinical settings, for patients with intracranial hemorrhage, the causes may be traumatic or spontaneous. Can these models apply to the differentiation of traumatic intracerebral hemorrhage or subarachnoid hemorrhage from hemorrhagic stroke? Can this study differentiate the perifocal edema of brain tumors from ischemic stroke? How can we use these models for any brain CT at emergency room? These critical points should be thoroughly discussed.

Response 4:  Images of trauma, trauma-related hemorrhage and brain tumor are classified into other findings in our study and the accuracy is about 0.98. So, we think this system can identify these different categories.

The clinical application of our result is mainly for a classifier for radiologists, who can quickly issue a warning to clinicians. Rather than a “red-dot system”, we will apply this result to further develop a system combined with nature language process (NLP). Although understanding the mechanism underlying the “black-box” in computer operation is difficult, we can supply a lots of image inputs for NLP training through this classifier. And that is what we are committed to do in the future.

Point 5:  Many sentences need further English editing (some short paragraphs should be merged.) to reconstruct the manuscript.

Response 5:  We will consider extensive English revision again from MPDI website if still needed. English editing certificate as following:

Round 2

Reviewer 1 Report

Thanks for your answers. However, I suggest adding more information about your answers in the document. 

Q3: If you have used a number of images between 24767*1 and 24767*2, please explain this in your paper

Q6.  You have illustrated a dense layer in Fig1 but not in fig 2 and 3. Consequently, I suggest saying that you have illustrated it in Fig. 1 and add your answer "" Our dense layers consist of 3 hidden layers, including 50,176 input units and 4 output units. Dense layers of VGG-16 and CNN-2 are equipped with rectified linear unit (ReLU) activation function, but the dense layer of Resnet-50 didn’t."

Q7. I have suggested adding an image for each category. If you've located a false positive image, you could add it even if you can't explain why the classification failed.

Author Response

Response to Reviewer 1 Comments

Point 1: Q3: If you have used a number of images between 24767*1 and 24767*2, please explain this in your paper

Response 1: We have added “We randomly applied each type of augmentation for each image. So, the range of our data size is about 1 to 2 folds of the original (24767*1~24767*2).” in our paper.

Point 2: Q6. You have illustrated a dense layer in Fig1 but not in fig 2 and 3. Consequently, I suggest saying that you have illustrated it in Fig. 1 and add your answer "" Our dense layers consist of 3 hidden layers, including 50,176 input units and 4 output units. Dense layers of VGG-16 and CNN-2 are equipped with rectified linear unit (ReLU) activation function, but the dense layer of Resnet-50 didn’t."

Response 2: We have added “A dense layer is also referred to as a fully-connected (FC) layer (illustrated in Fig.2 and Fig. 3). Our dense layers consist of 3 hidden layers, including 50,176 input units and 4 output units. Dense layers of VGG-16 and CNN-2 are equipped with rectified linear unit (ReLU) activation function, but the dense layer of Resnet-50 didn’t.” in Fig. 1

Point 3: Q7. I have suggested adding an image for each category. If you've located a false positive image, you could add it even if you can't explain why the classification failed.

Response 3: We add Fig. 6 in our discussion, which included false positive examples.

Reviewer 2 Report

Most of the questions and comments were answered and the manuscript was revised accordingly. I suggest to accept it for publication.

Author Response

Dear Prof.:
Thanks very much for your kind work and consideration on publication of our paper. On behalf of my co-authors, we would like to express our great appreciation to editor and reviewers.
Thank you and best regards.